# Nutrient-Enriched Germinated Brown Rice Alters the Intestinal Ecological Network by Regulating Lipid Metabolism in Rats

**DOI:** 10.3390/ijms26167693

**Published:** 2025-08-08

**Authors:** Chuanying Ren, Shuwen Lu, Shan Shan, Shan Zhang, Bin Hong, Di Yuan, Jingyi Zhang, Shiwei Gao, Qing Liu, Xiaobing Fan

**Affiliations:** 1Rice Processing Research Laboratory, Food Processing Research Institute, Heilongjiang Academy of Agricultural Sciences, Harbin 150086, China; chuanying1023@163.com (C.R.); 18845896856@163.com (S.S.); zhangshanfood@163.com (S.Z.); gru.hb@163.com (B.H.); yuandi199707@163.com (D.Y.); 18846080235@139.com (J.Z.); 2Rice Breeding Research Laboratory, Suihua Brunch of Heilongjiang Academy Agricultural Sciences, Suihua 152000, China; gaoshiwei1118@126.com (S.G.); liuqing58627@163.com (Q.L.); 3National Center of Technology Innoation, Salt-Alkali Tolerant Rice, Sanya 572000, China; fxb@lpht.com.cn

**Keywords:** germinated brown rice, nutrient rich, lipid metabolism, gut microbiota, related gene

## Abstract

Metabolic diseases such as high blood lipids, high blood sugar, and disrupted gut microbiota pose a serious threat to people’s physical health. The occurrence of these diseases is closely related to the lack of nutrients in daily rice staple foods, but there is a lack of comprehensive analysis of the underlying mechanisms. This study used fully nutritious brown rice as raw material, and after germination under various stress conditions, it significantly increased the levels of gamma aminobutyric acid (GABA, four carbon non protein amino acid), resistant starch, flavonoids, and other components that regulate metabolic diseases. Using rats as experimental subjects, a model of hyperlipidemia and hyperglycemia was constructed, with rice consumption as the control. The experimental period was 8 weeks. Research has found that feeding sprouted brown rice can significantly improve the accumulation of white fat in the liver caused by a high-fat diet, significantly reduce TC, TG, LDL-C, apoB, HL, LPL, and LCAT, significantly increase HDL-C and apoA1, and significantly reduce the levels of inflammatory factors IL-6 and TNF-α. Therefore, consuming sprouted brown rice can reduce the risk of hyperlipidemia, inflammation, and tumor occurrence by promoting fat breakdown, and can also increase the abundance of metabolic-promoting microorganisms (especially Euryarchaeota and Lactobacillus) in the intestine, improving the entire metabolic ecological network of rats.

## 1. Introduction

Rice is the staple food of half of the world’s population. Fine milled rice is the endosperm structure of rice, containing only carbohydrates and a small amount of protein, with significant loss of many nutrients. Brown rice is a whole grain with an intact embryo and husk [1]). Germinated brown rice enriched with various nutrients and bioactive substances is obtained via cultivation under optimal temperature, humidity, and oxygen conditions. Its components include γ-aminobutyric acid (GABA [2], γ-glutamylin (GO) [3], plant sterols [4], flavonoids [5], phenolic acids [6], vitamins [7], proteins and amino acids [8], carbohydrates [9], lipids [10], and minerals [11]. Compared to that in brown rice, cooked grain hardness is reduced by approximately 32% in germinated brown rice due to changes in the cortex microstructure and decreased insoluble dietary fiber levels [12]. Additionally, grain texture is soft and quality is significantly improved in germinated brown rice [13]. Germinated brown rice shows significantly reduced inorganic As content and improved safety [14]. It is a well-known functional whole grain widely used as a staple food and raw material for other foods [15]. It is also used as a probiotic carrier to produce functional value-added fermentation products [16].

Germinated brown rice, rich in the aforementioned compounds, significantly regulates human metabolism and intestinal health. It alleviates hyperglycemia, insulin resistance, and inflammatory insulin resistance induced by a high-fat diet and promotes the proliferation of probiotic species in the intestine of mice [17]. Germinated brown rice decreases the triglyceride (TG), total cholesterol (TC), low-density lipoprotein cholesterol (LDL-C), and apolipoprotein (apo)-B levels and increases the high-density lipoprotein cholesterol (HDL-C) and apoA1 levels, exerting potent lipid-lowering effects in patients with hyperlipidemia [18]. Germinated brown rice consumption improves the intestinal diversity by increasing the relative abundance of Megamonas and decreasing that of Veyrococcus in patients with type 2 diabetes and dyslipidemia [19].

Previous studies have demonstrated the correlation between germinated brown rice and regulation of functions related to sugar metabolism, lipid metabolism, and gut microbiota. However, specific interactions and action mechanisms of the entire metabolic ecological network remain unknown. Therefore, in this study, multi-stress germination technology was used to produce nutrient-enriched germinated brown rice, and its regulation of glucose and lipid metabolism and intestinal microbiota, metabolite, and related gene levels was comprehensively investigated using rats. We will comprehensively reveal the correlation and mechanism of nutrient-enriched germinated brown rice in regulating the entire metabolic ecological network of hyperlipidemia, hyperglycemia, inflammation, tumors, gut microbiota, and gut metabolites, and provide a theoretical basis for its future applications.

## 2. Results

### 2.1. Nutritional Components of Germinated Brown Rice

This study performed ultrasound and ozone treatments during brown rice germination and obtained nutrient-enriched germinated brown rice via high-temperature sterilization after germination. The representative characteristic components of germinated brown rice related to glucose and lipid metabolism are listed in Table 1. Here, the sample moisture content was 35.58%. All results obtained on a dry weight basis are indicated below.

Compared with brown rice, the characteristic component GABA in nutrient-enriched germinated brown rice increased by 14.57 times. During the germination process of brown rice, the action of proteases promotes the breakdown of proteins into amino acids. Compared with brown rice, the total amount of free amino acids in germinated brown rice increased by 4.7%. In addition to the activation effect of the enzymes mentioned above, a series of complex biochemical reactions also occur inside germinated brown rice, synergistically promoting the synthesis and accumulation of flavonoids, which is 2.51 times higher than that of brown rice. High temperature sterilization stress treatment not only increased the content of resistant starch, but also increased the insoluble dietary fiber content in germinated brown rice by 23.1% compared to brown rice.

### 2.2. Rat Liver Morphology

In this study, rat livers were collected for the analysis of morphological changes (Figure 1).

Hematoxylin and eosin staining revealed that the liver lobules in each rat group were intact, liver cell morphology was normal, and nuclei were clear, with no pathological changes, such as necrosis or degeneration. The clarity and uniformity of nuclei in the Gbrown group were better than those in the model and R-CK groups, indicating that germinated brown rice consumption protects the liver. The model group exhibited large lipid droplet vacuoles in the cytoplasm of liver cells, indicating severe steatosis and fatty liver. However, the Gbrown group exhibited significantly reduced and evenly distributed lipid droplet vacuoles in the cytoplasm, indicating that germinated brown rice consumption alleviated fatty liver formation in rats.

### 2.3. Blood Lipid Levels and Inflammation in Rats

Compared to the blank group, the model group showed significantly higher TC, TG, and LDL-C levels and significantly lower HDL-C levels due to the high-fat diet, confirming successful model establishment and severe hyperlipidemia. Feeding germinated brown rice during modeling significantly improved the four blood lipid indicators more in the Gbrown group than in the R-CK group (Figure 2a). This was because the high-fat diet significantly decreased the apoA1 levels (*p* < 0.05) and significantly increased the apoB levels (*p* < 0.05) in the model group, reducing their ability to transport cholesterol from the surrounding tissues to the liver for metabolism. Instead, excessive lipid transport occurred in extrahepatic tissues, resulting in increased blood lipid levels. However, germinated brown rice consumption significantly increased the apoA1 levels and significantly decreased the apo-B levels, thereby significantly improving hyperlipidemia in the Gbrown group more than that in the R-CK group (Figure 2b). Due to the high-fat diet, the model group showed significantly reduced hepatic lipase, lipoprotein lipase, and lecithin-cholesterol acyltransferase activities, weakening their fat-degrading ability. However, germinated brown rice consumption significantly increased the activities of all three lipases, thereby promoting fat breakdown and improving hyperlipidemia in the Gbrown group (Figure 2c). Compared to the blank group, the high-fat diet-fed model group exhibited increased interleukin (IL)-6 and tumor necrosis factor (TNF)-α levels, showing enhanced inflammation and tumor development. Consumption of germinated brown rice significantly reduced the IL-6 and TNF-α levels, thereby inhibiting inflammation and tumor development in the Gbrown group (Figure 2d).

### 2.4. Lipid Metabolism-Related Factors in Rats

Aspartate transaminase (AST) and alanine transaminase (ALT) are essential enzymes for sugar and protein interconversion in the human body. TBA is the final product of cholesterol breakdown and liver metabolism. It is often used in clinical practice to evaluate the functional status of the liver, and an increase in its value typically indicates liver dysfunction [20]. Compared to the blank group, the model group showed significantly elevated AST, ALT, and TBA levels due to the high-fat diet, which also caused liver damage. However, brown rice consumption significantly reduced the three indicators in the brown group, indicating the protective effects of brown rice against liver damage (Figure 3a,b). Glycated hemoglobin (GHb) is a combination of hemoglobin and carbohydrates in the serum; it is the gold standard to assess blood sugar control. GHb levels depend on the blood sugar levels and contact time between the blood sugar and hemoglobin but are unrelated to other factors, such as blood drawing time, fasting, and use of insulin. Increased GHb levels increase the risk of diabetic nephropathy and atherosclerosis. Compared to the blank group, the model group showed significantly higher GHb and blood sugar levels due to the high-fat diet. However, the Gbrown group showed significantly decreased GHb and blood sugar levels after brown rice consumption (Figure 3b).

Leptin (LEP) and adiponectin (ADPN) are protein hormones secreted by the adipose tissue. LEP enters the bloodstream, reduces food intake, and inhibits adipocyte synthesis, ultimately leading to weight loss [21]. DPN plays an important role in regulating insulin sensitivity and glucose metabolism [22]. Compared to the blank group, the model group showed significantly lower LEP and ADPN levels due to the high-fat diet. However, compared to the model group, the Gbrown group showed significantly higher LEP and ADPN levels after consuming germinated brown rice, which inhibited fat synthesis and regulated the blood glucose balance (Figure 3c). ET is a lipopolysaccharide stimulating monocytes and macrophages to produce inflammatory mediators, such as IL and TNF. Compared to the blank group, model group showed significantly higher ET levels due to the high-fat diet. In contrast, Gbrown group showed significantly decreased ET levels after consuming germinated brown rice, which reduced the risk of inflammation and tumor development (Figure 3d).

### 2.5. Rat Intestinal Microbiota

Principal component analysis visually displays the differences in the types and abundance of microorganisms and metabolites among groups. Blank group exhibited the least overlap with the model group, indicating the lowest similarity in the gut microbiota of the two groups. High-fat diet significantly altered the type and abundance of gut microbes. After consuming germinated brown rice, the Gbrown group approached the blank group with the largest overlap, indicating that the gut microbiota was regulated in this group (Figure 4a). The distance between the blank and model groups was the greatest, indicating no similarity in the horizontal microbiota of the two groups. High-fat diet caused significant changes in the type and abundance of gut microbes. After consuming the germinated brown rice, the Gbrown group was closer to the blank group, showing an overlap of approximately 50%, indicating significant improvements in the type and abundance of gut microbes (Figure 4b). Blank and model groups exhibited no overlap and were far apart, indicating that the high-fat diet completely altered the gut metabolites in these groups. Overlap between the Gbrown and blank groups reached approximately 80%, indicating a significant improvement in gut metabolites (Figure 4c).

At the phylum level, Actinobacteria and Euryarchaeota exhibited the highest abundances in the Gbrown group, accounting for 22.6% of all microbes. Cyanobacteria, Defirribaterota, and Desulfobacterota were more abundant in the model group, whereas Bacteria, Actinobacteria, and Verrucomicrobota exhibited the highest abundance in the R_CK group. Firmicutes exhibited relatively higher abundances in the model and blank groups; however, the blank group exhibited the highest abundance of Bacteroidota with a low Firmicutes/Bacteroidetes ratio (Figure 5a). At the genus level, *Lactobacillus* and *Bifidobacterium* are probiotics playing important roles in the prevention and treatment of high-cholesterol and cardiovascular diseases [23,24]. Abundance of probiotic *Lactobacillus* was the highest in the Gbrown and bank groups, accounting for 30.7 and 21.8% of all microbes, respectively. Abundance of probiotic *Bifidobacterium* was the highest in the Gbrown and R_CK groups, accounting for 22.6% of all microbes in the former group (Figure 5b). These results suggest that germinated brown rice consumption promotes probiotic proliferation.

### 2.6. Correlation Between Gut Microbiota and Metabolite Levels in Rats

At the phylum level, five microorganisms were significantly correlated with the top 20 differential metabolites, among which increased Euryarchaeota abundance showed significant positive correlations with the 20-carboxy-leukotriene B4, 2-indolecarboxylic acid, Tyr-Leu-Tyr, and Ile-Phe-Ala metabolites (Figure 6a). Metabolite 20-carboxy-leukotriene B4 binds to the BLT1 receptor with high affinity, and BLT-1 inhibits scavenger receptor class B member 1-mediated lipid transfer between HDL and cells [25]. Moreover, 5-chloroindole-2-carboxylic acid, a derivative of metabolite 2-indolecarboxylic acid, significantly reduces the plasma cholesterol levels in rats, with a potency twice that of chlorofibrate, without affecting the liver weight [26]. Analog 5-methoxyindole-2-carboxylic acid sustainably decreases the blood glucose levels in fasting rats [8]. In this study, germinated brown rice consumption promoted the proliferation of Euryarchaeota, which is related to glucose and lipid metabolism, and significantly increased the abundance of Actinobacterota, which promotes the breakdown of fat into short-chain fatty acids, which are beneficial for weight control [27].

At the genus level, an increase in the abundance of *Lactobacillus* was associated with the following metabolites: Sanguiin H1, 20-carboxy-leukotriene B4 L-threo-sphingosine, 2-indolecarboxylic acid, Tyr-Leu-Tyr, beta-tyrosine, and Ile-Phe-Ala. It was positively correlated with 1-docosanoyl-2-tetradecanoyl-sn-glycoro-3-phosphoryline and strongly negatively correlated with the metabolites, taxol B, nilotinib, isocouclidine, and Leu-Thr (Figure 6b). Increase in the abundance of *Bifidobacterium* was positively correlated with sanguiin H1, 20-carboxy-leukotriene B4, L-threo-sphingosine, 2-indolecarbboxylic acid, Tyr-Leu-Tyr, and Ile-Phe-Ala, and also positively correlated with amitrilose, nilotinib, 2-amino-4-hydroxy-6-pyropho-sphosphoryl-methylpteridine, nitroso (2S)-2-amino-4-sulfanylbutanoate, and Leu-Thr are negatively correlated [28].

### 2.7. Lipid Metabolism-Related Genes in Rats

Notably, 3-hydroxy-3-methyl-glutaryl-CoA reductase (*HMGCR*) controls the expression levels of cholesterol elimination-related genes, including the low-density lipoprotein receptor (*LDLR*), peroxisome proliferator-activated receptor alpha (*PPAR-α*), and liver X receptor alpha (*LXRA*). Some natural products accelerate the conversion of cholesterol to bile acid by regulating the sterol regulatory element-binding protein 2 (*SREBP2*) and *HMGCR* levels to control the bile acid level in the feces [29]. Compared to the blank group, the high-fat diet-fed model group showed significantly lower levels of cholesterol elimination-related genes (*PPAR-α*, *LXRA4*, and *HMGCR*) and significantly higher levels of cholesterol synthesis-related gene *SREBP2*. Compared to the model group, the Gbrown group exhibited significantly higher levels of cholesterol elimination-related genes (*PPAR-α*, *LXRA4*, and *HMGC*) and significantly lower levels of cholesterol synthesis-related gene *SREBP2*, which were significantly higher than those in the blank group (Figure 7a). Consumption of germinated brown rice mainly reduced the cholesterol levels by upregulating the cholesterol elimination-related gene levels and downregulating the cholesterol synthesis-related gene levels in the liver.

Alpha-2-HS-glycoprotein (*AHSG*) is a novel renal cell carcinoma biomarker gene promoting cancer cell proliferation by regulating the transforming growth factor-β signaling pathway [30]. Glucose-6-phosphatase catalytic subunit (*G6PC*) is a key gene involved in gluconeogenesis and glycogen breakdown during glucose metabolism. It is abnormally expressed in various cancers and involved in tumor proliferation and metastasis [31]. Phosphoenolpyruvate carboxykinase 1 (*PCK1*) is involved in various metabolic processes, including carbohydrate and lipid metabolism. Low *PCK1* expression reduces the plasma glucose, glycolysis, gluconeogenesis, and adipogenesis gene levels, TG and TC levels, and lipid accumulation [32]. Compared to the blank group, the high-fat diet-fed model group showed significantly higher *AHSG*, *G6PC*, *PCK1*, and plasma glucose levels, fat accumulation, and cancer risk. In contrast, the Gbrown group exhibited significantly lower *AHSG*, *G6PC*, and *PCK1* levels than the model group (Figure 7b). Consumption of germinated brown rice reduced risks of high-fat diet-associated hyperlipidemia, hyperglycemia, and cancer.

## 3. Discussion

### 3.1. Metabolic Regulatory Effects of GABA

Under adverse stress conditions, plants promote rapid cellular stress, including mechanical damage and low-temperature, high-temperature, low-oxygen, and salt stress, which alter the H^+^ concentration in the cytoplasm, activate various enzymes, and change the levels of bioactive substances [33]. This study used three stress techniques, ultrasound, ozone, and high-temperature sterilization, to promote the synthesis and accumulation of characteristic component GABA in germinated brown rice, reaching 14.57 times that of brown rice, and showing significant effects in regulating glucose and lipid metabolism. This is because the intake of GABA can reduce fasting blood glucose in high-fat obese mice, activate the protein kinase A (PKA) signaling pathway, thereby increasing fat breakdown, improving serum lipid profile and liver fat production [33]. Previous studies have found that 600 mg of GABA restores the vitality of beta cells, resists streptozotocin toxicity [34], and increases the glucagon, adrenaline, growth hormone, and cortisol responses to hypoglycemia by over two-fold. It is used to treat type 1 diabetes and its complications [35]. In addition, a GABA-rich adzuki bean diet promotes glycogen synthesis in the liver and downregulates the *SREBP1c* levels, thereby inhibiting TG and cholesterol synthesis and improving hyperglycemia in type 2 diabetes mellitus model mice [36]. Fermented tea increases the GABA levels and reduces the serum cholesterol, LEP, insulin, and fasting blood glucose levels [37]. Additionally, GABA decreases blood sugar levels by acting on hypothalamic ventromedial targets and inhibiting antiregulatory hormone release [38].

### 3.2. Metabolic Regulation of RS

After germination of brown rice, high-temperature sterilization stress causes starch gelatinization. After storage at room temperature for one week, the resistant starch (RS) content significantly increases to 5.6 times that of brown rice. RS in *Sorghum* effectively controls the body weight and adipose tissue quality by increasing the HDL-C levels and decreasing the TG, TC, and LDL-C levels [29]. Kudzu root RS reduces fat accumulation and stabilizes the gut microbiota by regulating fatty acid metabolism and the PPAR signaling pathway [39]. Corn RS promotes fat breakdown and gluconeogenesis in the liver by activating the AMP-activated protein kinase signaling pathway and accelerating energy consumption during the growth stage in broilers [40]. Rice RS significantly reduces the blood glucose, TC, TG, AST, ALT, and alkaline phosphatase levels and body weight, improves the intestinal microbiota abundance, and promotes short-chain fatty acid production in rats [41]. Therefore, the significant content of RS in germinated brown rice plays an important role in regulating blood lipids, blood glucose levels, and gut microbiota.

### 3.3. Metabolic Regulatory Effects of Flavonoids

Brown rice promotes flavonoid accumulation after germination. Flavonoids alter the metabolite levels by changing the microbial abundance. Gut microbiota metabolize flavonoids. The relationship between flavonoids and gut microbiota maintains the metabolic balance in the body [42]. Flavonoids inhibit fat production, promote fat breakdown and apoptosis in adipose tissue cells, and exhibit antioxidant and anti-inflammatory activities to promote animal health [25,43]. They also upregulate the blood glucose levels, serum insulin levels, pancreatic functions, lipid pathways, and proinflammatory cytokine levels in streptozotocin-induced diabetes model rats [44]. *Lithospermum* flavonoids improve glucose metabolism and lipid distribution, reduce the serum glucose, TC, and TG levels, improve the liver histopathological state, and exert antioxidant effects in diabetic rats [45].

### 3.4. Metabolic Regulatory Effects of Insoluble Dietary Fibers

The liver is responsible for various metabolic activities in the human body [46]. The liver plays an important role in the digestion, absorption, breakdown, synthesis, and transportation of lipids. Liver cells secrete bile, and bile acid salts in bile can promote the digestion and absorption of lipids [33]. Liver damage decreases the bile secretion capacity of liver cells. The liver maintains a relatively constant blood glucose concentration via glycogen synthesis, breakdown, and gluconeogenesis. However, severely impaired liver function decreases the synthesis, breakdown, and gluconeogenesis of liver glycogen, making it difficult to maintain the normal blood glucose concentration [47] (Hosseini Dastgerdi, Sharifi et al. 2021). The insoluble dietary fiber in germinated brown rice increased to 1.23 times that of brown rice, which significantly reduced and evenly distributed lipid droplet vacuoles in the liver cytoplasm, reduced fat accumulation, and also significantly increased the abundance of Actinobacteriota and Euryarchaeota in the intestine. Insoluble dietary fibers reduce the proportions of Firmicutes and Bacteroidetes, increase the abundance of probiotics (e.g., *Lactobacillus* and Bacteroidetes), reduce the abundance of cellulose-degrading bacteria (e.g., *Romboutsia* and *Clostridia*-UCG-014), promote liver health, and decrease lipid deposition, thereby positively impacting glucose and lipid metabolism [48]. Insoluble dietary fibers in barley affect the gut microbiota composition and abundance, promote short-chain fatty acid biosynthesis, and exert anti-obesity effects in mice [49]. Flaxseed insoluble dietary fibers reduce fat accumulation and enhance the blood lipid profiles, basal metabolism, gut microbiota, and short-chain fatty acid levels in the gut of obese mice [50]. Insoluble dietary fibers significantly reduce the fasting blood glucose, 2-h postprandial blood glucose, glycated hemoglobin, TC, TG, and LDL-C levels. Notably, insoluble dietary fibers are more effective than soluble dietary fibers in reducing the fasting blood glucose levels [51]. Therefore, the increase of insoluble dietary fiber in germinated brown rice plays an important role in regulating sugar metabolism, lipid metabolism, gut microbiota, and metabolites.

## 4. Materials and Methods

### 4.1. Materials

Preparation of nutrient-enriched germinated brown rice: Briefly, 10 kg of Suijing 309 brown rice was weighed, washed, soaked in 2–3 times its volume of water at room temperature (20–25 °C) for 4 h, and placed at 30 °C for 40 h for constant temperature germination. During this period, water was sprayed at 30 °C constant temperature every 2 h for 10s. After 12–30 h of germination, 30 kHz ultrasonic treatment was performed for 20 min. After 18–40 h of germination, a 15 g ozone treatment was performed. After germination, 120 °C high-temperature sterilization was performed for 30 min to obtain the nutrient-enriched germinated brown rice.

### 4.2. Experimental Animals

Forty male Sprague–Dawley rats (specific pathogen-free grade; 8 weeks old; 180–220 g) were obtained from Beijing Weitonglihua Experimental Animal Technology Co., Ltd. (Beijing, China) (animal production license number: SCXK (Beijing) 2021-0006).

### 4.3. Main Reagent

The main reagents in this experiment are shown in Table 2.

### 4.4. Main Instruments

The following instruments were used in this study: Multiskan FC ELISA reader (Thermo scientific, Shanghai, China), BX53 microscope (OLYMPUS Corporation Limited, Toyko, Japan), K5600 ultra micro spectrophotometer (Beijing Kaiao Technology Development Co., Ltd., Beijing, China), TC1000-G-Pro PCR instrument (Dalong Xingchuang Experimental Instrument (Beijing) Co., Ltd., Beijing, China), A-15907-2213 electrophoresis apparatus (Shanghai Tanon Life Science Co., Ltd., Shanghai, China), and Tanon-5200 Fully Automatic Luminescence Imager (Shanghai Tanon Life Science Co., Ltd., Shanghai, China).

### 4.5. Composition Determination

GABA: High-performance liquid chromatography was used to determine the GABA content of rice. Three parallel tests were performed to obtain the average values with the following test parameters: C18 chromatographic column (diameter: 3 µm; 250 mm × 4.6 mm), acetonitrile:sodium trihydrate (35:65) mobile phase, 1.0 mL/min flow rate, 30 °C column temperature, 436 nm detection wavelength, and three-time parallel determination [52].

Soluble dietary fiber: Soluble dietary fiber content was determined based on the GB 5009.88-2014 “National Food Safety Standard for Determination of Dietary Fiber in Foods,” including oligosaccharides and partially indigestible polysaccharides, in three parallel tests.

Resistant starch (RS): RS content was determined via spectrophotometry. Non-RS in the sample was hydrolyzed to glucose using α-pancreatic amylase and starch glucosidase, followed by ethanol addition and centrifugation to obtain RS granules. These granules were dissolved in potassium hydroxide solution and hydrolyzed into glucose using starch glucosidase. After reacting with glucose oxidase peroxidase to produce a colored complex, absorbance was measured at 510 nm, which was proportional to the RS content. Three parallel measurements were taken [53].

Flavonoids: High-performance liquid chromatography-mass spectrometry was used to determine the flavonoid content under the following conditions: Waters HSS T3 column (50 × 2.1 mm; 1.8 μm), mobile phase A (ultrapure water solution containing 0.1% formic acid), mobile phase B (acetonitrile solution containing 0.1% formic acid), 0.3 mL/min flow rate, 40 °C column temperature, and 2 μL injection volume. In total, 16 types of flavonoids were detected and measured in three parallel tests [54].

### 4.6. Feeding of Experimental Animals

Forty rats were divided into model (model), blank (blank), rice (R-CK), and germinated brown rice (Gbrown) groups, with 10 rats per group. Each cage holds 5 mice, and each group consists of 2 cages. The feeding environment was controlled using an independent air conditioning fan system, with good indoor ventilation, 20–22 °C room temperature, and 28–50% relative humidity. The laboratory environment was kept clean and hygienic, and the rats were provided free access to food and water.

On day 9, all rats were fed a normal basal diet for adaptive feeding and provided free access to food and water daily. After adaptive feeding, except for the model group, which was fed maintenance feed, the other groups were fed a high-fat diet for 28 d. After fasting for 12 h on the evening of day 27, streptozotocin (30 mg/kg) was intraperitoneally injected into all rat groups, except the blank group, on day 28. After 4 h, an equal volume of 20% glucose solution was injected. After fasting for 12 h at night, fasting blood glucose levels were measured on day 29, and postprandial blood glucose levels were measured 2 h after feeding. Blood glucose levels > 16.9 mmol/L confirmed the successful establishment of the model. Blank group was fed a maintenance feed, model group was fed a high-fat feed, R-CK group was fed a high-fat feed with 36% rice, and Gbrown group was fed a high-fat feed with 36% germinated brown rice for 28 d.

### 4.7. Sample Collection

After the experiment, blood samples were collected from the abdominal aorta of all rats. After centrifugation at 3500 rpm for 20 min at 4 °C, serum samples were extracted and frozen at −80 °C. Half of the liver was fixed with 4% paraformaldehyde, and the other half was frozen at −80 °C. Feces were frozen at −80 °C.

### 4.8. Serum Sample Evaluation

The operating method outlined in the relevant reagent kit listed in Table 1 was used for serum sample evaluation.

### 4.9. Hematoxylin and Eosin Staining

The liver is composed of multiple lobules, each consisting of liver cells, hepatic sinusoids, bile ducts, and central veins. Liver cells are the main functional cells of the liver, exhibiting various metabolic activities [55]. Hepatic sinusoids are microvessels within the liver responsible for blood filtration and exchange. The bile duct is the channel for bile excretion. The liver plays important roles in the digestion, absorption, decomposition, synthesis, and transportation of lipids. Liver cells secrete bile, and bile acid salts in bile promote the digestion and absorption of lipids. Liver damage decreases the bile secretion capacity of liver cells. Liver maintains a relatively constant blood glucose concentration via glycogen synthesis, breakdown, and gluconeogenesis. However, severely impaired liver function decreases the synthesis, breakdown, and gluconeogenesis of liver glycogen, making it difficult to maintain the normal blood glucose concentration. Approximately 4-micrometer-thick rat liver slices were baked at 60 °C for 30 min, dewaxed with xylene I, II, and III for 5 min each, hydrated with ethanol, stained with hematoxylin for 10 min, washed with different concentrations of ethanol for 5 min, stained with eosin for 2 min, and sealed with neutral gum. Subsequently, liver morphology changes were observed using an optical microscope, and images were collected.

### 4.10. Quantitative Polymerase Chain Reaction (qPCR) Analysis

The genes and sequences related to glucose and lipid metabolism are shown in Table 3.

RNA extraction: After grinding the rat liver in liquid nitrogen, 1 mL of TRIzol was added and thoroughly mixed. The mixture was left to stand at 18–25 °C for 5 min, followed by the addition of 200 μL of trichloromethane. The mixture was then reversed, left to stand at room temperature for 5 min, and centrifuged at 13,000 rpm for 15 min at 4 °C. The upper aqueous phase was transferred to a 1.5-mL centrifuge tube, and an equal volume of isopropanol was added and mixed repeatedly. The mixture was left to stand at room temperature for 10 min and centrifuged at 13,000 rpm for 10 min at 4 °C. The supernatant was discarded, and 1 mL of 75% ethanol was added. The centrifuge tube was gently shaken to suspend the precipitate, which was dissolved in 20 μL of diethylpyrocarbonate water and stored at −80 °C until use. Then, 1 μL of RNA was extracted, and its content and purity were measured using an ultra-micro spectrophotometer.

Reverse transcription: Briefly, 4 μL 5×Uni All-in-one SuperMix for qPCR, 1 μL gDNA Remove, and RNase-free Water in a total volume of 20 μL were used for PCR under the following conditions: 50 °C for 5 min, 85 °C for 2 min, and 4 °C +∞. The product was stored at −20 °C.

Real-time PCR: To quantify cDNA to 200 ng/μL, the reaction system consisted of 10 μL 2 × ChamQ Universal SYBR qPCR Master Mix with 0.4 μL Primer F, 0.4 μL Primer R, and 8.2 μL diethylpyrocarbonate water. Amplification was performed using a fluorescent quantitative PCR instrument.

### 4.11. Statistical Analysis

Statistical analyses were conducted using the SPSS v.26.0 software. Results are expressed as the mean ± standard deviation. Single-factor analysis of variance was used to compare multiple groups exhibiting normality and homogeneity of variance. Least significant difference *t*-test was used for pairwise comparisons between groups, and K independent sample non-parametric test was used to compare the groups not exhibiting normality and homogeneity of variance. Repeated-measures data were analyzed using repeated-measures analysis of variance. Statistical significance was set at *p* < 0.05.

## 5. Conclusions

In this study, ultrasound and ozone treatments were performed during brown rice germination, followed by high-temperature sterilization to obtain nutrient-enriched germinated brown rice with significantly high RS and insoluble dietary fiber levels. Notably, germinated brown rice consumption for eight weeks significantly improved white fat accumulation in the liver, hyperlipidemia, hyperglycemia, and inflammation in high-fat diet-fed rats. Additionally, germinated brown rice significantly altered the expression levels of glucose and lipid metabolism-related genes and proteins. It also increased the abundance of glucose and lipid metabolism-related gut microbiota, thereby regulating the gut metabolite composition and levels. Our results suggest that long-term consumption of germinated brown rice significantly improves the entire ecological network, including hyperlipidemia, hyperglycemia, liver functions, gut microbiota, and gut metabolite levels, highlighting the benefits and widespread application potential of germinated brown rice as a healthy staple food worldwide.

## Figures and Tables

**Figure 1 ijms-26-07693-f001:**
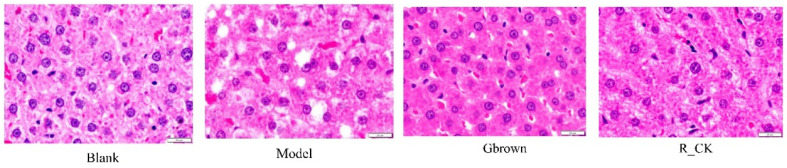
Hematoxylin and eosin (HE) staining of the rat liver tissues (400×). Blank, blank group; Model, model group; Gbrown, germinated brown rice group; R_CK, rice control group.

**Figure 2 ijms-26-07693-f002:**
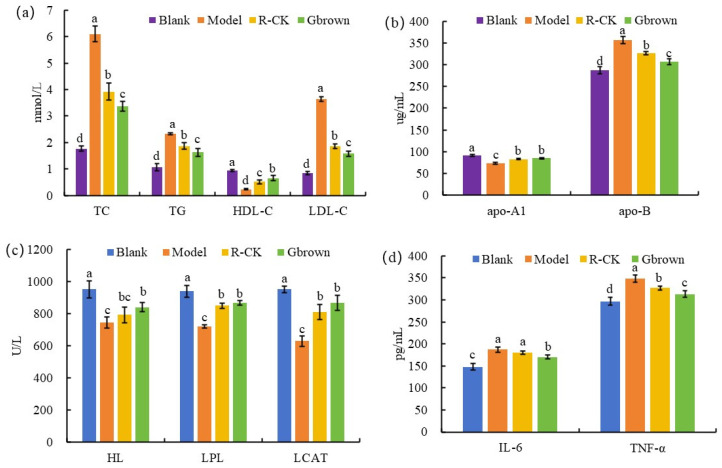
Blood lipid indicator and inflammatory factor levels. Blank, blank group; Model, model group; Gbrown, germinated brown rice group; R_CK, rice control group. (**a**) TC: cholesterol, TG: triglycerides, HDL-C: high density lipoprotein, LDL-C: low density lipoprotein. (**b**) apo-A1: apolipoprotein A1, apo-B: apolipoprotein B. (**c**) HL: hepatic lipase, LPL: lipoprotein lipase, LCAT: phosphatidyl cholesterol acyltransferase. (**d**) IL-6: interleukin-6, TNF-α: tumor necrosis factor alpha. In the figure, a, b, c, and d represent significant differences between groups (*p* < 0.05); number of samples *n* = 6.

**Figure 3 ijms-26-07693-f003:**
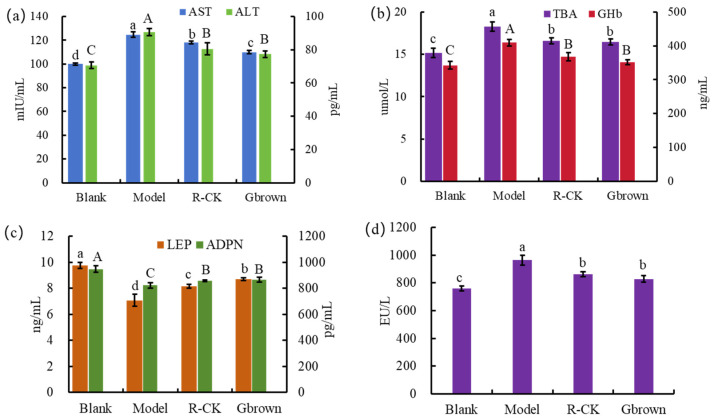
Levels of different glucose and lipid metabolism-related factors. Blank, blank group; Model, model group; Gbrown, germinated brown rice group; R_CK, rice control group. (**a**) AST: aspartate aminotransferase, ALT: alanine aminotransferase. (**b**) TAB: total bile acids, GHb: glycated hemoglobin. (**c**) LEP: leptin, ADPN: adiponectin. (**d**) ET: endotoxin. a, b, c, d or A, B, C, in the figure indicate significant differences between groups (*p* < 0.05) (*n* = 6/group).

**Figure 4 ijms-26-07693-f004:**
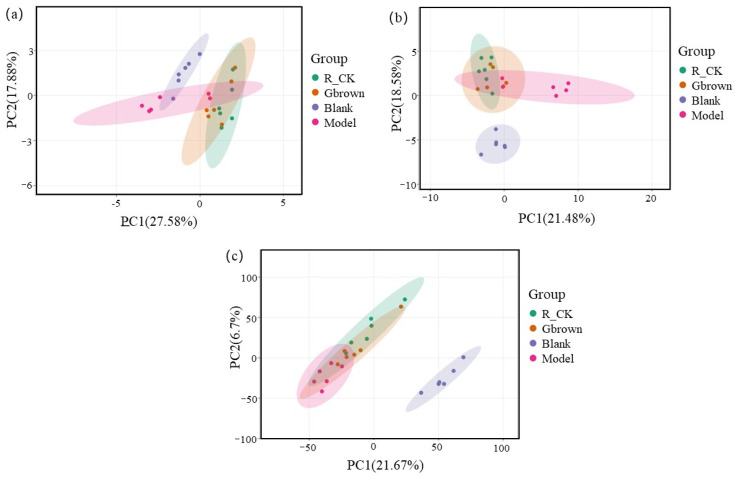
Principal component analysis (PCA) of gut microbiota and metabolite levels. Blank, blank group; Model, model group; Gbrown, germinated brown rice group; R_CK, rice control group. (**a**) Microorganisms at the phylum level. (**b**) Microorganisms at the genus level. (**c**) Metabolites. Different colors represent different groups, and points of the same color represent different samples within a group. The closer the distance between samples, the more similar the microbial composition between samples (*n* = 6/group).

**Figure 5 ijms-26-07693-f005:**
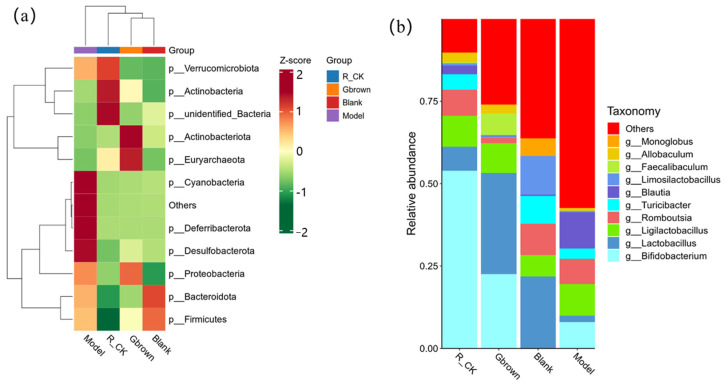
Relative abundances of the top 10 species at the phylum and genus levels in different groups. Blank, blank group; Model, model group; Gbrown, germinated brown rice group; R_CK, rice control group. (**a**) Clustering heatmaps, the horizontal axis represents groups and the vertical axis represents species. The closer to blue, the lower the abundance, and the closer to red, the higher the abundance. (**b**) Stacking bar charts for relative abundance. The horizontal axis represents groups, and the vertical axis represents relative abundance (*n* = 6/group).

**Figure 6 ijms-26-07693-f006:**
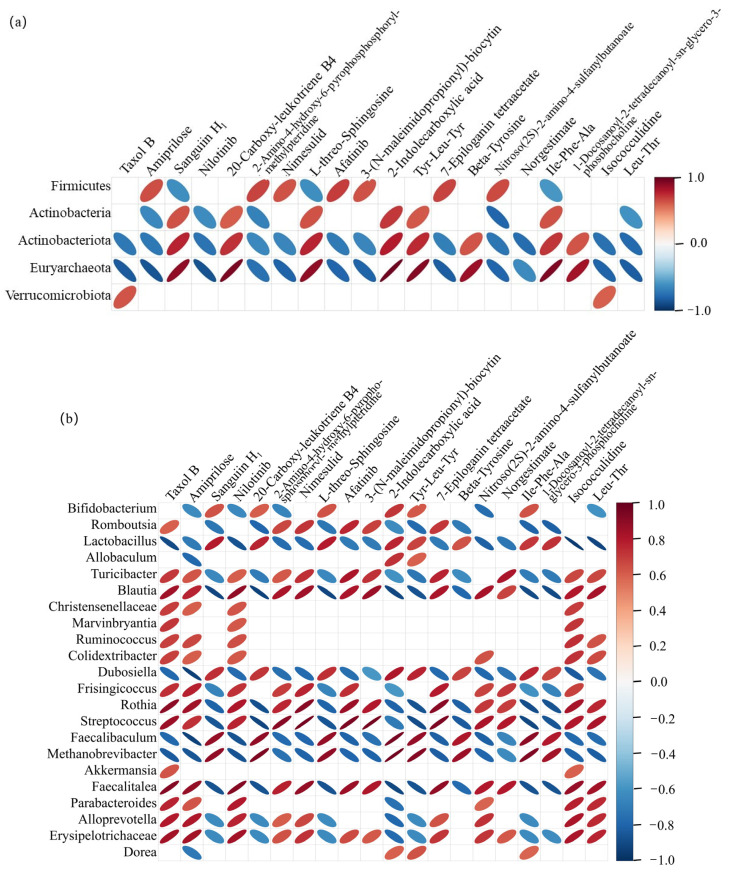
Spearman’s correlation heatmap of the top 20 differential metabolites and microorganisms. Blank, blank group; Model, model group; Gbrown, germinated brown rice group; R_CK, rice control group. (**a**) Microorganisms at the phylum level. (**b**) Microorganisms at the genus level. Behavioral microorganisms are classified as metabolites. The red ellipse represents positive correlation, while the blue ellipse represents negative correlation. The larger the absolute value of correlation, the finer the ellipse. Blank squares indicate a significance *p*-value greater than 0.05 (*n* = 6/group).

**Figure 7 ijms-26-07693-f007:**
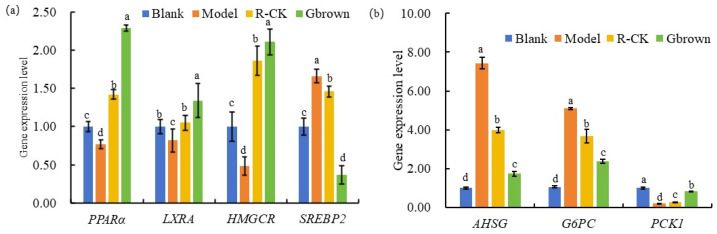
Expression levels of different genes and proteins related to lipid (**a**) and glucose (**b**) metabolism. Blank, blank group; Model, model group; Gbrown, germinated brown rice group; R_CK, rice control group. (**a**) Expression of genes related to lipid metabolism. (**b**) Expression of genes related to glucose metabolism. *PPARα*: peroxisome proliferator-activated receptor α mRNA; *LXRA*: recombinant liver X receptorα mRNA; *HMGCR*: 3-Hydroxy-3-methylglutaryl-CoA reductase mRNA; *SREBP2*: sterol regulatory element binding protein 2 mRNA; *AHSG*: recombinant alpha-2-Heremans Schmid glycoprotein mRNA; *G6PC*: glucose-6-phosphatase mRNA; *PCK1*: phosphoenolpyruvate carboxykinase 1 mRNA. a, b, c, d in the figure indicate significant differences between groups (*p* < 0.05) (*n* = 6/group).

**Table 1 ijms-26-07693-t001:** Nutritional components of germinated brown rice.

	GABA, mg/100g	Resistant Starch, %	Insoluble Dietary Fiber, %	Flavonoids, mg/100 g
Brown rice	2.38 ± 0.06	0.52 ± 0.02	0.78 ± 0.02	4.56 ± 0.08
Germinated brown rice	34.68 ± 0.08	2.89 ± 0.04	0.96 ± 0.03	11.44 ± 0.10

**Table 2 ijms-26-07693-t002:** Main reagents.

Reagents Name	Company	Location
Aspartate aminotransferase (AST) test kit	Shanghai Enzyme Linked Biotechnology Co., Ltd.	Shanghai China
Alanine aminotransferase (ALT) test kit		
Total Bile Acid (TBA) test kit		
Apolipoprotein A1 (apo-A1) test kit		
Apolipoprotein B (apo-B) test kit		
Glycated Hemoglobin (GHb) test kit		
Leptin (LEP) test kit		
Lipoprotein esterase (LPL) test kit		
Liver lipase (HL) test kit		
Egg Phosphocholesterol Acyltransferase (LCAT) test kit		
Adiponectin (ADPN) test kit		
Endotoxin (ET) test kit		
Interleukin-6 (IL-6) test kit		
Tumor necrosis factor alpha (TNF-α) test kit		
Total cholesterol (TC) test kit	Nanjing Jiancheng Technology Co., Ltd.	Nanjing China
Triglyceride (TG) test kit		
Low density lipoprotein cholesterol (LDL-C) test kit		
High density lipoprotein cholesterol (HDL-C) test kit		

**Table 3 ijms-26-07693-t003:** Genes and sequences related to glucose and lipid metabolism.

	Gene	Primer Sequence 5′-3′	Length (bp)
Lipid metabolism-related genes	*GAPDH*	F: TCTCTGCTCCTCCCTGTTCTA	121
		R: GGTAACCAGGCGTCCGATAC	
	*HMG* *CR*	F: ACTGAAACACGGGCATTGGGTT	195
		R: AACACGGCACGGAAAGAACCAT	
	*SREBP2*	F: GAGGCGGACAACACACAATA	370
		R: CGGCTCAGAGTCAATGGAATAG	
	*PPARα*	F: TCTGAACATTGGCGTTCGCA	98
		R: TCCCTCAAGGGGACAACCAG	
	*LXRA*	F: CTGCAACGGAGTTGTGGAAG	295
		R: TCGCAGCTCAGCACATTGTA	
Glucose metabolism-related genes	*GAPDH*	F: GGTGGACCTCATGGCCTACAT	84
		R: CTCTCTTGCTCTCAGTATCCTTGCT	
	*PCK1*	F: GGGTGGAAAGTTGAATGTGTGGGT	139
		R: TGGCGTTCGGATTTGTCTTC	
	*G6PC1*	F: AGCCTCTTCAAAAACCTGGGGA	110
		R: AAACGGAATGGGAGCGACTT	
	*AHSG*	F: ACATCTTCTTCAGGGATTCAGGCA	116
		R: AGGTTGTCTCCAGCGTGTCAAT	

GAPDH (glyceraldehyde 3-phosphate dehydrogenase, internal reference genes); HMGCR (3-hydroxy-3-methylglutaryl-coenzyme A reductase); SREBP2 (sterol-responsive element binding protein-2); PPARα (peroxisome proliferation-activated receptor-α); LXRA (liver X receptor α); *PCK1* (phosphoenolpyruvate carboxykinase 1); *G6PC1* (glucose-6-phosphatase 1); *AHSG* (alpha-2-HSglycoprotein).

## Data Availability

The original contributions presented in the study are included in the article. Further inquiries can be directed to the corresponding author.

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
