# Peer review of "Nutrient-Enriched Germinated Brown Rice Alters the Intestinal Ecological Network by Regulating Lipid Metabolism in Rats"

_ijms, 2025, doi:10.3390/ijms26167693_

Round 1

Reviewer 1 Report

Comments and Suggestions for Authors

Question about methods

What does model group mean?  Where rat ten to a cage or sing

e.  We’re rat weighs  and appearance recorded and if they were what was the result?  Wa s food consumption monitored?  Were any changes in appearance of gastrointestinal tract noted? Did signs of diarrhea orconstipation noted? I was confused about the  the timing  of treatment a table or figure would be he

llpful. Why was streazatocin administered?

Author Response

Please refer to the uploaded file for a detailed response. Thank you very much.

Reviewer 2 Report

Comments and Suggestions for Authors

This study aimed to evaluate the effects of nutrient-enriched germinated brown rice on lipid metabolism and the intestinal ecological network in rats with diet-induced metabolic disorders. The manuscript has several promising aspects, but it requires revisions as several points need to be carefully revised before the next resubmission, as follows:

  • The experimental design involves comparisons among more than two groups (Blank, Model, R-CK, Gbrown), making one-way ANOVA the appropriate statistical test. The use of multiple pairwise LSD t-tests increases the risk of Type I error. Please apply one-way ANOVA followed by a suitable post hoc test (e.g., Tukey’s) for multiple comparisons, and revise the figure legends and methods accordingly.
  • The abstract should be revised to include a brief background, a clear objective, and a short description of the experimental design. Add a concise conclusion to highlight the main findings and their relevance.
  • L38-39; The sentence “It is the endosperm structure of rice. It contains only carbohydrates and a small amount of protein…” is unclear. Please revise it.
  • The specific hypothesis of the study should be clearly stated before the objective at the end of the introduction.
  • The results section includes explanations and citations that are more appropriate for the Introduction or Discussion. Please remove them or move them to the Discussion section to maintain focus on the results only.
  • Lines 86-95; the text does not clearly explain or summarize the key findings in Table 3. Please explicitly describe the increases in GABA, resistant starch, flavonoids, and insoluble dietary fiber in germinated brown rice compared to untreated brown rice, and integrate these comparisons clearly into the narrative.
  • L90-91; The statement "GABA... exhibited a maximum increase of 14.57 times" compared to what baseline? Brown rice? This should be stated directly.
  • L97-108; these descriptions are basic liver physiology and not relevant to the results. These should be moved to the Introduction or Methods, or cut entirely.
  • In all figure legends, please define all abbreviations used. Additionally, clearly describe the experimental groups, indicate the number of replicates (n = ?/ group), and explain the meaning of significance letters or symbols used in the figures.
  • Overall, the Discussion section reads more like a literature review than a critical discussion of your own findings. Please focus on interpreting and contextualizing your experimental results rather than summarizing external studies.
  • Table 2: Add accession number or reference for each primer.

Author Response

(The authors gave the same response as above.)

Reviewer 3 Report

Comments and Suggestions for Authors

The manuscript of “Nutrient-enriched germinated brown rice alters the intestinal ecological network by regulating lipid metabolism in rats” by Ren Chuan-Ying and co-authors aimed to evaluate the effects of ultrasound/ozone-treated germinated brown rice obtained by multi-stress germination technology on glucose and lipid metabolism indices, intestinal microbiota, metabolite levels, and related gene expression levels in male Sprague–Dawley rats. The authors first found that 8-week consumption of germinated brown rice, which is significantly rich in resistant starch and insoluble dietary fiber, improved hyperlipidemia, hyperglycemia, liver functions, gut microbiota, and gut metabolite levels. The authors identified novel molecular mechanisms by which nutrient-enriched germinated brown rice can regulate the entire metabolic ecological network, providing a solid theoretical basis for its future applications.

Overall, the investigation was performed at a high experimental level. The study design was well-thought-out. The manuscript is well-written and contributes to the understanding of key pathways regulating gene expression altered by high-fat diet by germinated brown rice, demonstrating the advantages of germinated brown rice as a promising functional food with high sensory quality and nutritional value.

The relevance of the research topic is due to the sharp increase in the number of diabetes and metabolic disorders throughout the world and the urgent need to find new targets for their effective management. This study provides new genetic insights into the complex regulatory effects of germinated brown rice exposed to multifactorial stress in a high-fat diet/low-dose streptozotocin rat model, which is a widely used experimental model to mimic type 2 diabetes in animals.

The Title, Abstract, and Keywords correspond to the content of the manuscript. The Introduction section fully reflects the current state of the issue under study. The purpose of the study is clearly formulated in the Introduction. The authors' assumptions and conclusions are sufficiently substantiated. The manuscript contains a lot of interesting data, but some results are not presented clearly enough.

Comments:

A diagram or table outlining the study design should be provided at the beginning of the Results section to describe the interventions delivered to the experimental groups and the research methods used. Abbreviations used should be defined at the first mention.

The Results section is replete with known data on the indicators under study and references. This section could be significantly improved to avoid repetition with the Discussion section.

The authors need to describe in more detail the method of processing rice for improving its properties, first used in this work. It is unclear whether the choice of rice processing regimes was based on literature data or the authors' own research.

Lines 413-419: The percentage of fat in the high-fat diet used, as well as the method of preparation of the feed (or the catalog number of the commercial feed) were not indicated. Date on fasting blood glucose measured on day 29 and postprandial blood glucose measured 2 h after feeding were not presented.

Author Response

(The authors gave the same response as above.)

Round 2

Reviewer 1 Report

Comments and Suggestions for Authors

Instead of rats method stated fiv e mice in cage